



# Modeling fuel-, vehicle type-, and age-specific CO₂ emissions from global on-road vehicles, 1970-2020

Liu Yan[1], Qiang Zhang[2], Kebin He[1], Bo Zheng[3]

[1] State Key Joint Laboratory of Environment Simulation and Pollution Control, School of Environment, Tsinghua University, Beijing, People's Republic of China

[2] Ministry of Education Key Laboratory for Earth System Modeling, Department of Earth System Science, Tsinghua University, Beijing, People's Republic of China

[3] Institute of Environment and Ecology, Tsinghua Shenzhen International Graduate School, Tsinghua University, Shenzhen, China

*Correspondence to*: Qiang Zhang (qiangzhang@tsinghua.edu.cn)

**Abstract.** Vehicles are among the most important contributors to global anthropogenic $CO_2$ emissions. However, the lack of fuel-, vehicle type-, and age-specific information about global on-road $CO_2$ emissions in existing datasets, which are available only at the sector level, makes these datasets insufficient to support the establishment of emission mitigation strategies. Thus, a fleet turnover model is developed in this study, and $CO_2$ emissions from global on-road vehicles from 1970 to 2020 are estimated for each country. Here, we analyze the evolution of the global vehicle stock over 50 years, identify the dominant emission contributors by vehicle and fuel type, and further characterize the age distribution of on-road $CO_2$ emissions. We find that trucks accounted for less than 5% of global vehicle ownership but represented more than 20% of on-road $CO_2$ emissions in 2020. The contribution of diesel vehicles to global on-road $CO_2$ emissions doubled during the 1970-2020 period, driven by the shift in the fuel-type distribution of vehicle ownership. The proportion of $CO_2$ emissions from vehicles in developing countries such as China and India in terms of global emissions from newly registered vehicles significantly increased after 2000, but global $CO_2$ emissions from vehicles that survived more than 15 years in 2020 still originated mainly from developed countries such as the United States and countries in the European Union.



## 1 Introduction


To meet the Paris Agreement's 1.5℃ long-term temperature goal, many efforts have been made to
determine pathways for reducing the emissions of greenhouse gases such as $CO_2$ (Matthews & Caldeira,
2008; Meinshausen et al., 2009; Rogelj et al., 2018; Davis et al., 2018). Historical emission data and
consistent emission series of on-road vehicles, which are key sources of $CO_2$ emissions, are important
inputs for Earth system models, atmospheric chemistry and transport models, and integrated assessment
models to support studies on both climate change and global climate governance (Bhalla et al., 2014;
Janssens-Maenhout et al., 2019; Lelieveld et al., 2015; Niklas et al., 2020; Shindell et al., 2011; Silva et
al., 2016; Unger et al., 2010). Thus, estimating long-term $CO_2$ emissions from global on-road vehicles
with detailed source information is necessary as deep greenhouse gas emission reductions are pursued.
Several global emission inventories that cover emissions from on-road vehicles have been
developed and are widely used in global research and modeling. $CO_2$ emissions from on-road vehicles
can be derived from global anthropogenic emission inventories, including the Emissions Database for
Global Atmospheric Research (EDGAR), the Open-source Data Inventory for Atmospheric $CO_2$
(ODIAC), the Carbon Emission and Accounts Datasets (CEADs), and the Peking University (PKU)-$CO_2$
inventory. On-road $CO_2$ emissions are estimated with the total fuel consumption of the road sector at the
country level and fleet average emission factors in EDGAR (Amstel et al., 1999; Crippa et al., 2016;
Crippa et al., 2018; Janssens-Maenhout et al., 2019). Following the method in EDGAR, local data sources
are introduced more often in ODIAC (Boden et al., 2016; Boden et al., 2017; Od et al., 2018), CEDS
(Hoesly et al., 2018) and PKU-$CO_2$ (Wang et al., 2013) when estimating on-road $CO_2$ emissions. Global
$CO_2$ emissions from on-road vehicles in these widely used emission inventories are estimated as a whole
at the sector level in each country using the fuel-based method, and fleet structure information (e.g., fuel-,
vehicle type-, and age-specific characteristics) on on-road $CO_2$ emissions is omitted. Technology-based
models such as the Greenhouse Gas and Air Pollution Interactions and Synergies (GAINS) (Klimont et
al., 2017) and Speciated Pollutant Emissions Wizard (SPEW)-Trend (Tami et al., 2004 and 2007; Yan et
al., 2011 and 2014) models can be used to describe fleet structure information on emissions from global
on-road vehicles, but emission inventories built on these models include only emissions of air pollutants.
Here, a new global inventory of fuel-, vehicle type-, and age-specific $CO_2$ emissions from on-road
vehicles for each country from 1970 to 2020 is developed with the global fleet turnover model, in which



six types of fuel, five types of vehicles, and 231 countries are considered. Based on this inventory, we
analyze the evolution of the global vehicle stock over 50 years; identify the dominant emission
contributors by vehicle and fuel type; and further characterize the age distribution of on-road $CO_2$
emissions.

## 2 Materials and methods

### 2.1 Methodological framework

For a given country $c$, the annual $CO_2$ emissions from on-road vehicles in year $y$ are estimated as
follows:
$Emis_{c,y,v,f} = \sum_{i=0}^{i=T} Stock_{c,y,v} \times X_{c,y,v,i} \times FuelR_{c,y,v,f} \times VKT_{c,y,v,f} \times FE_{c,y,v,f} \times EF_{c,f},$     (1)
$Stock_{c,y,v} = V_{c,y,v}^* \times e^{\alpha_{c,v} e^{\beta_{c,v} E_{c,y}}} \times Population_{c,y},$     (2)
$Stock_{c,y,v} = \sum_{i=0}^{i=T} Sale_{c,y-i,v} \times Surv_{c,v,i},$     (3)
$X_{c,y,v,i} = Sale_{c,y-i,v} \times Surv_{c,v,i} / \sum_{i=0}^{i=T} Sale_{c,y-i,v} \times Surv_{c,v,i},$     (4)
$Fuel_{c,y,f} = \sum_{v} Stock_{c,y,v} \times FuelR_{c,y,v,f} \times VKT_{c,y,v,f} \times FE_{c,y,v,f},$     (5)
where $y$ is the target year, which ranges from 1970 to 2020; $i$ is the age of the vehicles registered in
year $(y-i)$; $T$ is the lifetime of vehicles; $v$ is the vehicle type, which includes two types of light-
duty vehicles, namely, passenger cars (PLDVs) and light commercial vehicles (CLDVs), two types of
heavy-duty vehicles, namely, buses and trucks, and motorcycles (MCs); and $f$ is the fuel type, which
includes gasoline, diesel, natural gas (NG), liquefied petroleum gas (LPG), electricity, and other fuels.
As shown in Eq. 1, annual $CO_2$ emissions ($Emis_{c,y,v,f}$) are estimated by the vehicle stock ($Stock_{c,y,v}$),
the fleet-average fuel structure ($FuelR_{c,y,v,f}$), the annual average kilometers traveled ($VKT_{c,y,v,f}$), the
fleet-average fuel economy ($FE_{c,y,v,f}$), the age distribution of the vehicle stock ($X_{c,y,v,i}$), and the $CO_2$
emission factor ($EF_{c,f}$). $Stock_{c,y,v}$ can be modeled using the Gompertz function (Eq. 2), which is an S-
shaped curve determined by two negative parameters ($\alpha$ and $\beta$), with the saturated vehicle stock per
1000 people ($V^*$), per capita GDP ($E$), and population ($Population_{c,y}$) as inputs. The age distribution of
the vehicle stock ($X_{c,y,v,i}$), which represents the proportion of surviving vehicles registered in year
$(y-i)$ in target year $y$, is modeled on the basis of the dynamic balance function (Eqs. 3 and 4) using
the number of newly registered vehicles ($Sale_{c,y-i,v}$) and survival rates ($Surv_{c,v,i}$). Fuel consumption by
vehicle type, which is calculated using $Stock_{c,y,v}$, $X_{c,y,v,i}$, $FuelR_{c,y,v,f}$, $VKT_{c,y,v,f}$, and $FE_{c,y,v,f}$, is
constrained by total on-road fuel consumption ($Fuel_{c,y,f}$) at the country level (Eq. 5).

84        In this study, the fleet turnover emission model (Figure 1) is constructed based on functions 1-5. In
summary, we first build an integrated vehicle stock database by combining and harmonizing the available
vehicle stock data from a series of global, regional and national statistics and filling data gaps with the
modeled stock based on the Gompertz function (Eq. 2). Second, the age distribution of the stock is
simulated with a combined vehicle sale statistical database and an integrated vehicle stock database using





the dynamic balance function (Eqs. 3 and 4). Then, vehicular fuel consumption is estimated using outputs
from the first two steps and other vehicle activity-related data and is constrained by national fuel
consumption statistics (Eq. 5). Finally, fuel- and vehicle type-specific $CO_2$ emissions from global on-
road vehicles from 1970 to 2020 are modeled on the basis of constrained vehicular fuel consumption and
$CO_2$ emission factors (Eq. 1).

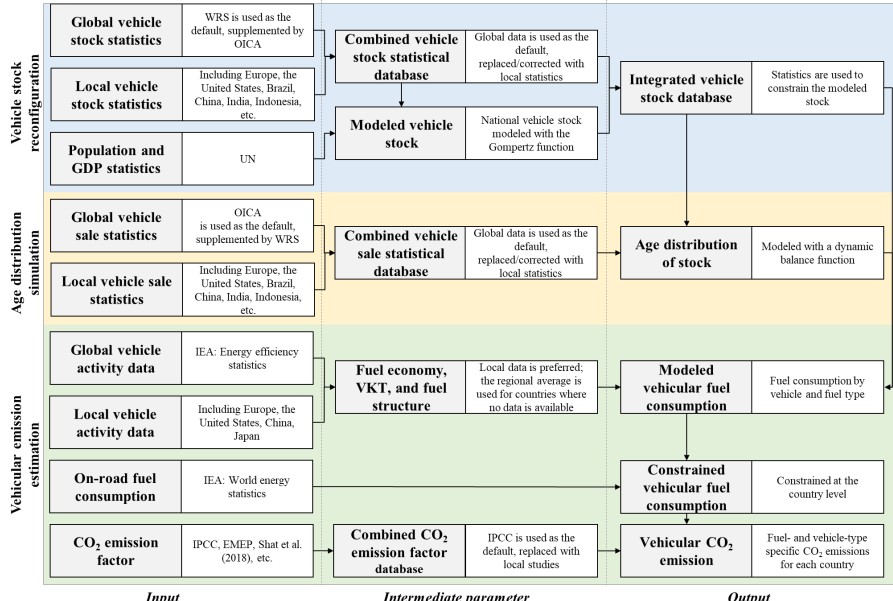


Fig. 1. Schematic methodology for estimating vehicular $CO_2$ emissions.
**2.2 Modeling the vehicle stock**
In the first step, an integrated vehicle stock database from 1970 to 2020 was constructed with both
statistical and modeled data. The statistical data used in this study was collected from various available
vehicle stock statistics, in which global statistics were used as the default vehicle stock and local statistics
were used to supplement and amend the default data. When statistical data was unavailable for a country
in a given year, vehicle stock modeled by the Gompertz function was used.
To determine the default vehicle stock database, two widely used vehicle stock statistics from the
Wold Road Statistics (WRS) 2021 Edition (IRF) and the International Organization of Motor Vehicle
Manufacturers (OICA) were collected and compared. We found that the trends of vehicle stock in the
WRS and OICA data were similar, but the absolute value of the vehicle stock in the OICA data was lower
than that in the WRS data, especially for developing countries (Figure S2). Taking India as an example,
the vehicle stock in the OICA data was 85% less than that in the WRS data. To further confirm the
reliability of these two global databases, local statistics were used for comparison. The WRS data were
more similar to the local vehicle statistics than were the OICA data (Figure S2). After comprehensive
consideration of spatiotemporal coverage, updating frequency and stability, and data reliability, the WRS
data were used as the default for global vehicle statistics, and the OICA data were used if there were no
data available from the WRS.
We also collected a series of local statistics as supplements and amendments to the global vehicle



statistics, in which 49 developing and developed countries were included (ACEA; CEIC; EC; JAMA;
MEIC; MOSPI; NBS; TEDB). By coupling multiple global and local vehicle databases, a combined
vehicle statistical database by vehicle category was established in this study. As the division of vehicle
types varied among statistics, we established a mapping relationship of vehicle types between this study
and other data sources (Table S2).
Given that statistical data of vehicle was unavailable before 2000 for most countries, the Gompertz
function, which was often applied to establish the relationship between vehicle ownership and an
economic indicator (Dargay and Gately, 1999; Dargay et al., 2007; Huo and Wang, 2012), was
subsequently used in this study to model the vehicle stock. In this study, per capita GDP was calculated
with national GDP (NBS; UNdata; WB) and population (NBS; WPP) as the economic indicator. The
saturated vehicle stock per 1000 people was first derived from previous studies (Huo and Wang, 2012)
and then adjusted by the maximal vehicle stock per 1000 people calculated using statistical data. The
combined vehicle statistical database was used to estimate parameters ($\alpha$ and $\beta$) of the Gompertz
function at the country level. For countries whose $R$ square ($R^2$) of the country-level regression was less
than 0.5, regional or global $\alpha$ and $\beta$ regression parameters were used instead (Zheng et al., 2012).
As the verification of the vehicle stock modeled by the Gompertz function, we compared them with
the statistical vehicle stock for countries in years when statistics were available. The relative deviation
ratios in countries that own top 85% of global vehicles stock were between -28% and 25.6%, ranges of
the relative deviation in rest countries were a bit larger due to the limited availability of statistics. Figure
2(a) and Figure S3 show the comparison in 2015, a year with more statistical data. The deviation of the
modeled vehicle stock from the statistics in most countries was less than $\pm25\%$, especially in the United
States, countries in the European Union, China, and India. The relatively good consistency between the
modeled and statistical vehicle stock indicates the relatively high reliability of this model. Therefore, a
long-term integrated vehicle stock database (1970-2020) was constructed by constraining the modeled
vehicle stock by the combined vehicle statistical database.

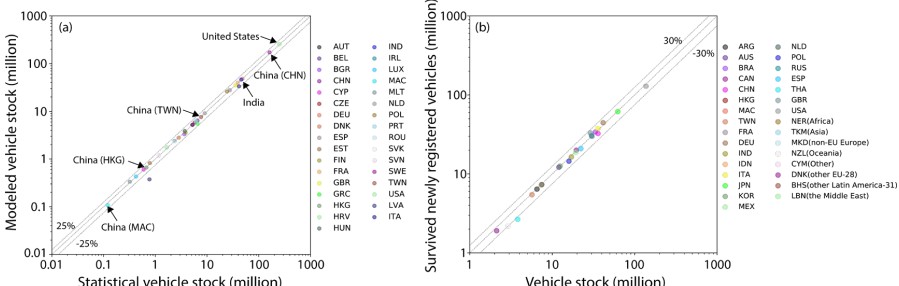


Fig. 2. Verification of the modeled vehicle stock in United States, the European Union, China, and India
(a) and the age distribution for PLDVs (b) in 2015.
**2.3 Modeling the age distribution of vehicle stock**
Then, the age distribution of the stock was modeled using the dynamic balanced function with the
integrated vehicle stock database set up in the first step and a combined vehicle sale statistical database.
Similar to the combination of vehicle stock statistics, OICA data were used as the default vehicle sale
database with WRS data as a supplement after comparison, and local statistics (ACEA; CEIC; EC; JAMA;
MEIC; NBS; TEDB) were also involved to correct the default database. Limited by the temporal
coverage of the statistical data, vehicle sales were not available for most countries before 2005. Therefore,



the newly registered vehicles for missing years was back-calculated with the dynamic balanced function,
in which the vehicle stock from the previous step and survival rates derived from available studies and
reports (Huo and Wang 2012; Yan et al., 2011; Yan et al., 2014; Zheng et al., 2014) were inputs. Here we
marked 231 countries into two types: focus countries and broader regions (Table S1). 20 countries
owning the top 75% of global vehicles were marked as focus countries, for which the dynamic balanced
function was built at country level. The remaining 211 countries were marked as broader regions and
further combined into 8 regions according to the roadmap region definition (ICCT 2012). In each broader
region, data in a reprehensive country, which has most abundant statistics with region, was used to build
the dynamic balanced function and the age distribution in this country was assumed to be able to represent
that in other countries belonging to the same region. The age distribution in this study was not simulated
for MCs due to the limitation of data availability, and we assumed that they shared the same age
distribution of PLDVs.
To verify the age distribution modeled by the dynamic balanced function, relative deviation between
the simulated vehicle stock based on newly registered vehicles and survival rates and the vehicle stock
in the first step was used as the validation indicator. Except for several years in Argentina and Thailand,
the relative deviation ratios of light-duty vehicles during 1970-2020 ranges from -30.9% to 30.8%,
heavy-duty vehicles had larger relative deviation ratios which were between -36.5% and 34.9%. Taking
2015 as an example, the relative deviation ratios in most countries were less than $\pm 30\%$ (Figure 2(b)
and Figure S4). The relatively good consistency between the vehicle stock and simulation indicated that
the dynamic balance function set up in this study could well model the entry of newly registered vehicles
and the retirement of existing vehicles and the estimated age distribution was reliable.
**2.4 Estimates of fuel consumption**
In the third step, we estimated the initial vehicular fuel consumption based on outputs from the first two
steps and parameters including the annual average kilometers traveled (VKT), fuel structure, and fuel
economy. Then the initial vehicular fuel consumption was constrained with energy statistics from World
Energy Statistics (IEA[1]) at country level, which was finally used in $CO_2$ estimation. VKT, fuel structure,
and fuel economy are rarely available in global statistics annually, this study used fleet-average data,
which were estimated based on vehicle-kilometers, the vehicle stock, vehicle-kilometer energy intensity,
and fuel consumption by category in energy efficiency statistics (IEA[2]). These indexes for 39 countries
(accounting for 43%-73% of the global vehicle stock) during the 2000-2018 period can be found in
energy efficiency statistics. For countries that were not covered in energy efficiency statistics, the
regional or global mean VKT, fuel structure, and fuel economy were used. For missing years, we assumed
that the values of these three parameters were similar to those of the adjacent year. There are few local
statistics or studies that evaluate the VKT, fuel structure, and fuel economy; therefore, these parameters
were supplemented and revised only for the United States, Europe, China, and Japan using local statistics
or studies (AECA; IEA[3]; JAMA; MEIC; TEDB; TRACCS).
As the validation of fuel consumption, the initial vehicular fuel consumption was compared to
energy statistics by fuel type (Figure S5). The range of relative deviation ratios of gasoline, diesel, NG,
and LPG was -23% to 3%, -19% to 9%, -22% to 34%, and -39% to 14%, respectively. As $CO_2$ is not
directly emitted as exhaust by electrical vehicles whether they were running, starting or parking,
electricity was not considered in the estimation of vehicular fuel consumption in this study. The
consistency of the simulation with statistics ensured the feasibility of constraining the modeled fuel
consumption by statistics.

**2.5 Estimates of CO$_2$ emissions and uncertainty assessment**

Finally, vehicular CO$_2$ emissions were estimated using the constrained vehicular fuel consumption from previous step and a combined CO$_2$ emission factor database in which emission factors from the Intergovernmental Panel on Climate Change (IPCC) were used as the default emission factors, and local studies (EEA; Shan et al., 2018) were used as supplements and amendments. As the CO$_2$ emission factor is influenced mainly by the fuel type and country, the estimation of CO$_2$ emissions would not be interfered with by the simplified assumption for MCs in modelling the age distribution.

Following the method in Crippa et al. (2018) and Crippa et al. (2019), the corresponding uncertainty ($\sigma$) of CO$_2$ emissions from on-road vehicles in year $y$ for a given country $c$ is calculated as following:

$$\sigma_{Emis_{c,y}} = \sqrt{\Sigma_f \left( \sigma^2_{AD_{c,y,f}} + \sigma^2_{EF_{c,f}} \right) \times \left( Emis_{c,y,f}/Emis_{c,y} \right)^2} \tag{6}$$

where $\sigma_{AD}$ and $\sigma_{EF}$ are the uncertainties (%) of the activity data (the constrained fuel consumption of on-road vehicles) and CO$_2$ emission factors. Based on assumption of lognormal distribution of the calculated uncertainties (Bond et al., 2004), we evaluated the upper and lower range of CO$_2$ estimate by multiplying and dividing the base emissions in this study by $(1 + \sigma)$, respectively (Crippa et al., 2018).

As CO$_2$ uncertainty can vary significantly among countries (Marland et al., 1999; Olivier et al., 2014) and the primary source of uncertainty of the CO$_2$ estimate from on-road vehicles is the activity data rather than emission factors (GPG 2000), the main step in CO$_2$ uncertainty assessment is to evaluate the uncertainty of national activity data. In this study, 231 countries were divided into several groups (Table S1) in the uncertainty assessment in accordance with IPCC tiered approach and EDGAR (Janssens-Maenhout et al., 2019). Here we assume that countries belonging to the OECD in 1990 (OECD90) have the lowest uncertainties in their fuel consumption data because they were economically stable and would have a good statistical infrastructure. On the same line, fuel consumption data in countries with Economies in Transition of 1990 (EIT90) is more uncertain than that of OECD90 but less than that from the other remaining non-Annex I countries. Exceptions to the country grouping are made for Australia, Canada, China, India, Japan, Russia, Ukraine, United States, and countries belonging to the 15 member countries of European Union (EU15) whose uncertainty values of fuel consumption data were obtained from Olivier et al. (2016) and Hong et al. (2017). Uncertainty values for CO$_2$ emission factors were retrieved from EEA.

Table S4 shows the corresponding uncertainty of CO$_2$ emissions at both global and regional level during 1970-2020 on basis of Eq. 6. The uncertainty in the global on-road CO$_2$ emissions is estimated to range from -7.2% to 8.1%, which is close to the expert judgement suggested value (approximately $\pm5\%$) in GPG (2000). Because sufficient local data was used in the CO$_2$ estimation, United States and European Union have the lowest uncertainty in the range of -3.8% to 4.0% and -2.9% to 3.0%, respectively. India also has relatively low uncertainty that varies between -4.7% and 5.0% because of the low uncertainty derived from Janssens-Maenhout et al. (2019) in which India is classified as countries with well-developed statistical systems. Due to the less-developed statistical systems, Latin Am. + Canada and Middle East + Africa have the largest uncertainty, which range from -12.3% to 14.6% and -15.4% to 18.3%, respectively. Hong et al. (2017) found that the apparent uncertainties in oil consumption during 1996-2003 were relatively large with an average apparent uncertainty ratio of 15.8%, which led to the relatively larger uncertainty in China's on-road CO$_2$ emissions with the range of -12.6% to 14.4%. It could also be found that uncertainties at regional level decreased over time with the development of statistical systems in more countries. But uncertainty in global on-road CO$_2$ emissions slightly increased



during 1970-2020 due to the growing contribution of regions with larger uncertainty to the global total
$CO_2$ emissions.
**3 Results**
**3.1 Evolution of the global vehicle stock, 1970-2020**
The global vehicle stock continuously increased from 0.3 billion in 1970 to 2.3 billion in 2020, and there
is both consistency and variety between countries in terms of the distributions of vehicles and fuel types
(Figures 3 and S7). In 1970, PLDVs were the major vehicle type in United States (83%) and the European
Union (88%) but had relatively low proportions in China (23%) and India (5%). The high proportion of
PLDVs in the United States and the European Union, as well as the dominant position of these two
regions in terms of the global vehicle stock (Figure S6), led to more than 70% of global vehicles being
PLDVs in 1970. The proportion of PDLVs in China significantly increased and reached 68% in 2020 and
have replaced MCs to become the dominant vehicle type. Although the stock of PLDVs in India also
increased substantially during the 1970-2020 period, MCs were still the most frequently used vehicles,
accounting for 78% of the vehicle stock in India in 2020. In 2020, the majority of vehicles in the European
Union were still PLDVs, for which the proportion was 79%, but the dominant vehicle type in United
States has changed from PLDVs to CLDVs, which accounted for 50% of the local vehicle stock. With
the replacement of developed countries by developing countries in terms of the global vehicle stock
during the 1970-2020 period (Figure S6), the share of MCs in the global vehicle stock increased
accordingly to 32%, and the proportion of PLDVs decreased to 50% in 2020.
Unlike the changes in the vehicle-type distribution during the 1970-2020 period, the fuel structure
of the vehicle stock was consistent in most regions. Currently, the majority of the vehicle stock worldwide
still consists of gasoline and diesel vehicles, which together accounted for 98% of the global vehicle
stock in 2020. Gasoline was the major fuel type for vehicles in most countries from 1970 to 2020, but
the dieselization of PLDVs in regions such as the European Union (Figure S10) led to a larger proportion
of diesel vehicles in the local vehicle stock. For example, the share of diesel vehicles in the European
Union increased from 29% in 1970 to 43% in 2020. Although the share of electrical vehicles in the
vehicle stock was still much lower than that of gasoline and diesel vehicles, the stock of global electrical
PLDVs has reached 10.2 million, and in this regard, the growth has been the fastest in the last eight years.

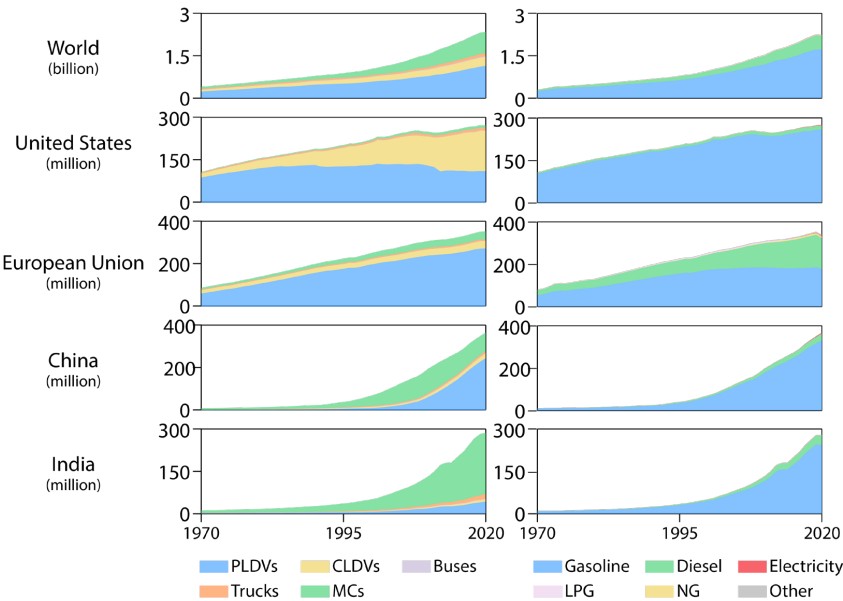


Fig. 3. Trends in vehicle ownership from 1970 to 2020.
**3.2 CO$_2$ emissions from global on-road vehicles**
Global CO$_2$ emissions from on-road vehicles continued to increase overall from 1.7 Gt in 1970 to 5.4 Gt
in 2020 (Figure 4). Profiting from the integrated global vehicle database developed in this study, we
further analyzed the vehicle- and fuel type-specific characteristics of CO$_2$ emissions from global on-road
vehicles. On-road CO$_2$ emissions were concentrated in specific vehicle and fuel types throughout the
period. From 1970 to 2020, almost all of global CO$_2$ emissions from on-road vehicles came from gasoline
and diesel vehicles due to their dominant proportion in the vehicle stock (Figure S10). In 1970, 78% and
21.5% of global on-road CO$_2$ emissions were exhausted from gasoline and diesel vehicles, respectively,
and in 2020, these emissions together accounted for 96% of global on-road CO$_2$ emissions; only the
ranking of the contributions changed. With continuous dieselization during the 1970-2020 period (Figure
S10), the contribution of diesel vehicles to global on-road CO$_2$ emissions increased to 47% in 2020.
Although CO$_2$ emissions from vehicles using other fuels (here, NG and LPG) continued to grow during
the 1970-2020 period, their proportions were still quite slight compared to those of gasoline and diesel
vehicles.
PLDVs, accounting for the largest share in the global vehicle stock, were also the main source of
global on-road CO$_2$ emissions and contributed more than 47% of global CO$_2$ emissions from on-road
vehicles during the 1970-2020 period. Although MCs accounted for the second largest share in the global
vehicle stock, CO$_2$ emissions from MCs were not comparable to those from PLDVs. In 2020, proportion
of PLDVs and MCs in the global vehicle stock was 50% and 32%, respectively, and their CO$_2$ emissions
were 2.6 Gt and 0.3 Gt, respectively, which accounted for 48% and 5% of global on-road CO$_2$ emissions,
respectively. In contrast, trucks with a fairly low share in the global vehicle stock contributed the second
largest share of global on-road CO$_2$ emissions. During the 1970-2020 period, trucks accounted for less
than 5% of the global vehicle stock but exhausted 17% of global on-road CO$_2$ emissions in 1970, and
their contribution increased to 22% in 2020. As most PLDVs are gasoline vehicles and the majority of
trucks are powered by diesel, gasoline PLDVs and diesel trucks are among the top 2 vehicle- and fuel
type-specific contributors to global on-road $CO_2$ emissions. In 2020, the $CO_2$ emissions from gasoline
PLDVs and diesel trucks were 1.8 Gt and 1.1 Gt, respectively, accounting for 33% and 20% of global
on-road $CO_2$ emissions, respectively.

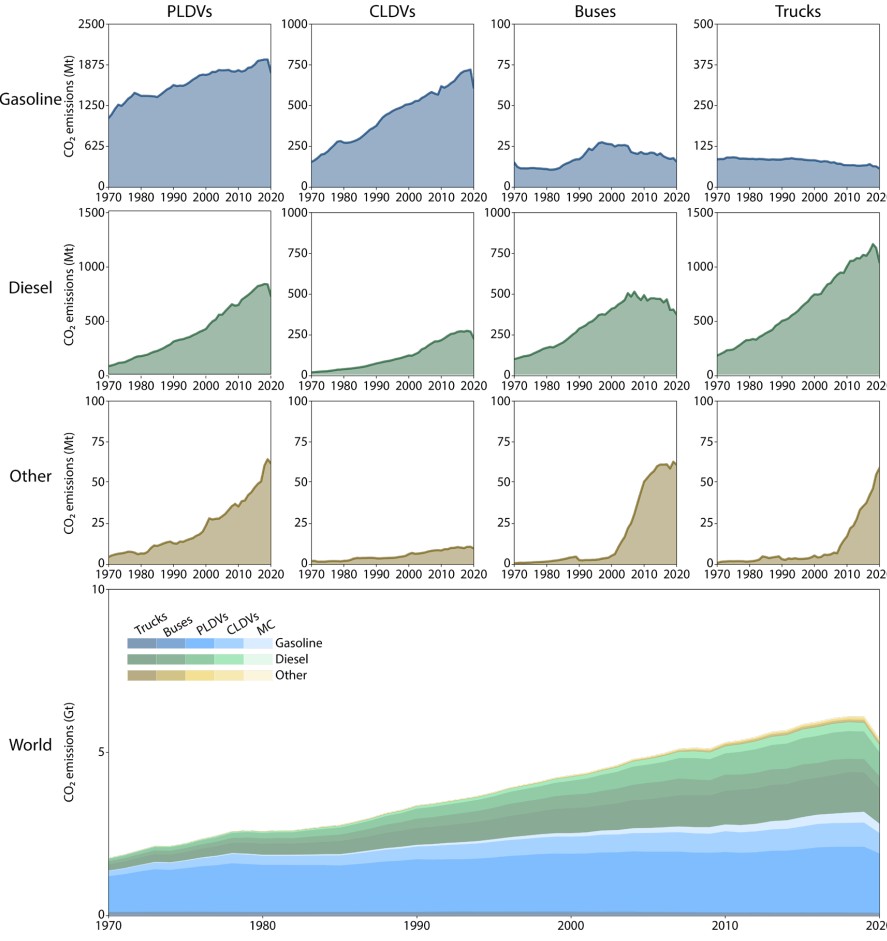


Fig. 4. Global $CO_2$ emissions from 1970 to 2020 by vehicle and fuel type. The panels are organized by
fuel type (rows) and vehicle type (columns)
Figure 5 shows the geographical distribution of the two largest contributors to global on-road $CO_2$
emissions in 2020, namely, gasoline PLDVs and diesel trucks. Global on-road $CO_2$ emissions were highly
concentrated in several countries. In 2020, the top 10 countries contributed 69% and 71% of global $CO_2$
emissions exhausted from gasoline PLDVs and diesel trucks, respectively. The United States was still
the largest contributor to global $CO_2$ emissions from both gasoline PLDVs and diesel trucks, whose
contributions were up to 25% and 28%, respectively. With the continuous improvement in China's
economic development, China became the leading market for global vehicles in 2020 (Figure S6) and
accounted for 18% and 19% of $CO_2$ emissions from global gasoline PLDVs and diesel trucks,
respectively. Although growth in on-road $CO_2$ emissions in developed countries slowed down after 2000
(Figure S8), the contributions of gasoline PLDVs and diesel trucks in developed countries were still
greater than those in developing countries, especially for gasoline PLDVs. For example, the ownership
of gasoline PLDVs in Canada and India was relatively close in 2020, at 22.5 and 21.2 million,
respectively, but the $CO_2$ emissions from gasoline PLDVs in Canada were 83.5 Mt, which is three times
greater than that in India.

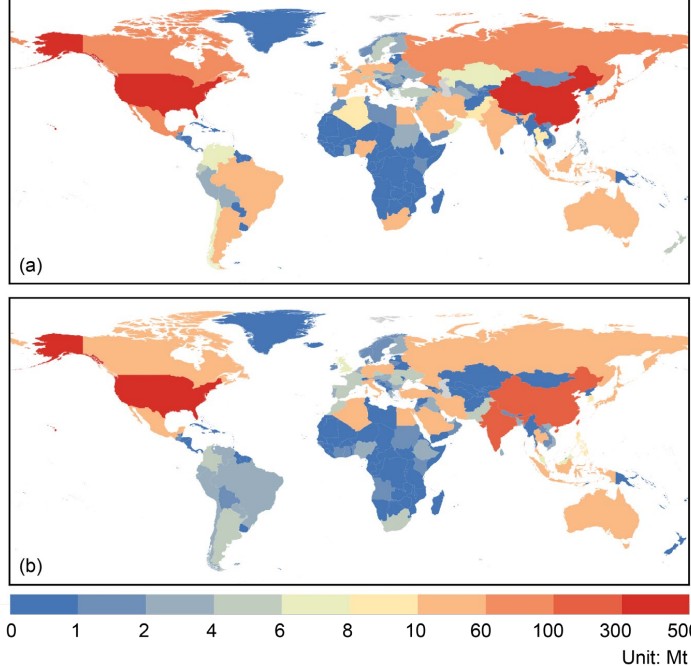


Fig. 5. Maps of on-road $CO_2$ emissions from the top 2 contributors worldwide: (a) gasoline PLDVs and
(b) diesel trucks.
We further analyzed the influence of shifts in the fuel-type distribution of vehicle ownership (Figure
S10) on the fuel structure of on-road $CO_2$ emissions (Figure 6 and Figure S11). In 1970, $CO_2$ emissions
from PLDVs were mainly exhausted from gasoline vehicles, as the majority of PLDVs in most regions
were powered by gasoline, and diesel vehicles exhausted only 7% of $CO_2$ emissions from PLDVs
worldwide. In 2020, gasoline vehicles were still the dominant contributor to $CO_2$ emissions from PLDVs
in the United States and China, but the contribution of diesel vehicles increased significantly in the
European Union and India, which accounted for 61% and 50% of local $CO_2$ emissions from PLDVs,
respectively. Influenced by the dieselization of PLDVs in regions such as the European Union and India,
the contribution of diesel vehicles to $CO_2$ emissions from PLDVs in 2020 also increased to 28%. For
CLDVs, the contribution of diesel vehicles was more than 50% in the European Union, China, and India,
but in the remaining regions, $CO_2$ emissions were still mainly from gasoline vehicles. Buses and trucks
were also dieselized during the 1970-2020 period, and diesel vehicles have become the dominant
contributor to $CO_2$ emissions from buses and trucks both regionally and globally. Therefore, controlling
emissions from diesel vehicles, especially buses and trucks, holds great significance for reducing global
on-road $CO_2$ emissions.

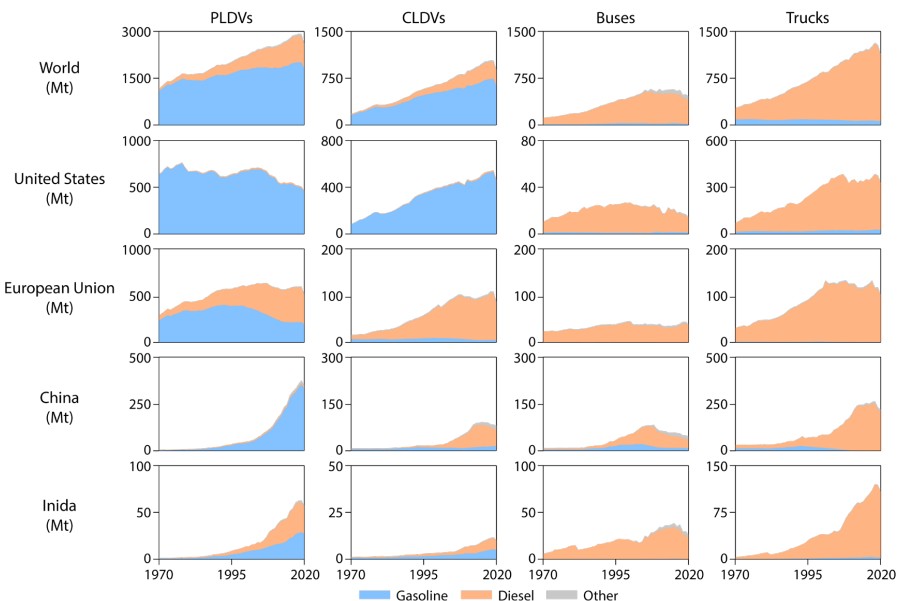


Fig. 6. Transition of diesel vehicles' contribution to $CO_2$ emissions.
**3.3 Age distribution of $CO_2$ emissions**
On the basis of the fleet turnover emission model built in this study, the age distribution of global on-
road $CO_2$ emissions was estimated and analyzed (Figure 7). The contribution of old vehicles (those that
survived more than 15 years) to $CO_2$ emissions was relatively low, regardless of whether they were light-
duty or heavy-duty vehicles. In 1970, old vehicles contributed 4% and 6% of $CO_2$ emissions from light-
duty and heavy-duty vehicles, respectively. Although the contribution of old vehicles to $CO_2$ emissions
increased, they still contributed only approximately 10% of $CO_2$ emissions from both light-duty and
heavy-duty vehicles in 2020. As emissions of air pollutants such as particulate matter (PM) may increase
with age because of degradation in engine performance and air pollution control equipment (Yan et al.,
2011), the contributions of old vehicles to emissions of air pollutants could be much greater than those
of $CO_2$. Therefore, controlling old vehicles may not be significant in mitigating $CO_2$ emissions but could
lead to effective air pollutant emission coreductions.
Global $CO_2$ emissions from vehicles of all ages were mainly contributed by developed countries,
such as the United States and countries in the European Union before 2000, as these countries owned the
majority of global vehicles during that period. After 2000, the contributions of vehicles in developing
countries such as China and India to global on-road $CO_2$ emissions increased significantly, especially for
$CO_2$ emissions from vehicles younger than ten years. Taking $CO_2$ emissions from light-duty vehicles
aged 0-1 as an example, the proportion of these vehicles in China increased from 1% in 1970 to 16% in
2020, while the proportion of these vehicles in the United States decreased from 44% in 1970 to 23% in
2020. $CO_2$ emissions from old vehicles in 2020 were still mainly exhausted by vehicles in developed
countries such as the United States and countries in the European Union, which is related to the longer
lifetimes and earlier development of vehicles in these countries. For example, old vehicles in the United
States contributed nearly half of the $CO_2$ emissions exhausted from old light-duty vehicles worldwide in
352     2020.

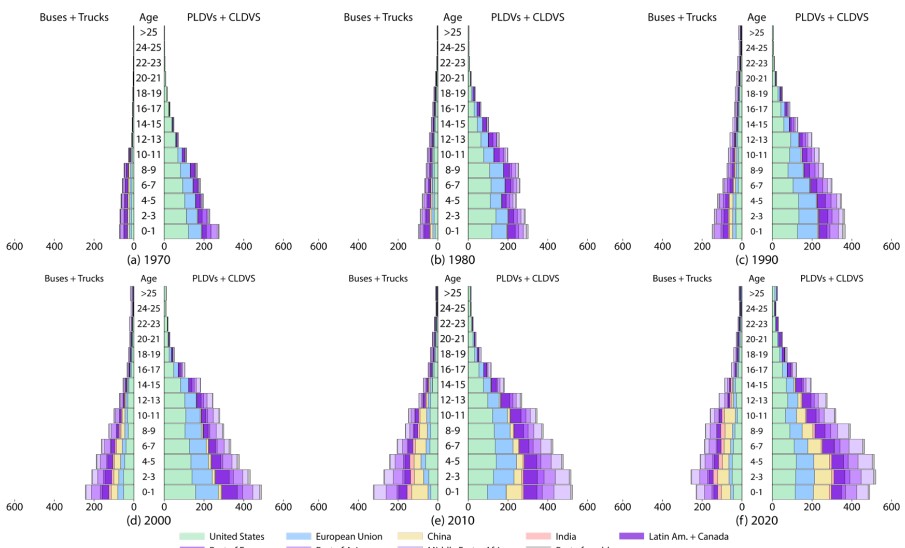

Fig 7. Shares of $CO_2$ emissions by vehicle age. In each panel, the bars from left to right show the proportions of the world, the United States (US), the European Union (EU), China, and India accounted for by vehicles in the vehicle age categories. The panels are organized by year (rows) and vehicle type (columns).

## 4 Data availability

The fuel-, vehicle type-, and age-specific $CO_2$ emission data presented herein cover the period from 1970 to 2020 at the country level. The data are available as open data at https://doi.org/10.6084/m9.figshare.24548008.v5 (Yan et al., 2023).

## 5 Conclusions

Our study constructed a fuel-, vehicle type-, and age-specific $CO_2$ emission inventory from 1970 to 2020 of global on-road vehicles covering 231 countries, five types of fuel, and five types of vehicles. In this model, the best available statistics on the vehicle stock and sales were used to model the vehicle stock via the Gompertz function as well as the age distribution based on the dynamic balanced relationship between the vehicle stock and vehicle sales. Statistical fuel consumption was used to constrain the estimated vehicular fuel consumption at the country level, and emission factors from both the IPCC and local studies were used to estimate $CO_2$ emissions. On the basis of our $CO_2$ emission inventory with detailed information, the evolution of the global vehicle stock over 50 years was analyzed, the dominant emission contributors by vehicle and fuel type were identified, and the age distribution of on-road $CO_2$ emissions was also characterized. We found that trucks accounted for less than 5% of global vehicle ownership but represented more than 20% of on-road $CO_2$ emissions in 2020. The contribution of diesel vehicles to global on-road $CO_2$ emissions doubled during the 1970-2020 period, driven by the shift in the fuel-type distribution of vehicle ownership. The proportion of $CO_2$ emissions from vehicles in developing countries such as China and India in terms of global emissions from newly registered vehicles

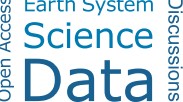

significantly increased after 2000, but global $CO_2$ emissions from vehicles that survived more than 15
years in 2020 still originated mainly from developed countries such as the United States and countries in
the European Union.
The fleet turnover model built in this study could also be used for estimating global on-road
emissions of air pollutants, which are more significantly influenced by the vehicle-type distribution, fuel
structure, and age distribution of the fleet. However, these fuel-, vehicle type-, and age-specific
characteristics have not yet been discussed in existing studies. In the future, our model could help
improve the global emission inventory of air pollutants from on-road vehicles and further support
analyses of coreductions in $CO_2$ and air pollutant emissions from global on-road vehicles as well as the
potential air quality and climate cobenefits. In addition to the uncertainty quantification for our $CO_2$
emission data, we further verified the reliability of $CO_2$ emissions in this study by comparing them to
those of other widely used global, regional, and national emission inventories in which long-term $CO_2$
emissions are available (Figure S12). The $CO_2$ emissions in this study not only exhibited good
consistency with other global emission inventories at the global scale but also were more similar to local
emissions than those in other global or regional emission inventories at the country and regional levels.
**Supplement.** The data related to figures in this article is available in the supplementary file Figures.zip.
**Author contributions.** LY collected the data, developed the fleet turnover model, and constructed the
database of fuel-, vehicle type-, and age-specific $CO_2$ emissions from global on-road vehicles during the
1970-2020 period. LY and QZ discussed the expansion of the database. LY wrote the paper with the help
of all the coauthors.
**Competing interests.** At least one of the (co-)authors is a member of the editorial board of Earth System
Science Data.
**Acknowledgments.** This work was supported by the National Natural Science Foundation of China
(41921005, 42130708, and 72140003).

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
