# Peer review of "Modeling fuel-, vehicle type-, and age-specific CO2"

_Earth System Science Data, 2024_

## Author Response (AR1)

**RC1**

**General comments:**

--In the introduction part, several global emission inventories are mentioned. I propose to describe in more detail the problems with current inventories and the motivation of this work.

More detail has been added to the description of the motivation of this work as well as the limitation of current inventories. The last paragraph of the introduction has been changed as:

Here, a new global inventory of fuel-, vehicle type-, and age-specific $CO_2$ emissions from on-road vehicles for each country from 1970 to 2020 is developed with the global fleet turnover model, in which six types of fuel, five types of vehicles, and 231 countries are considered. Based on this inventory, we analyze the evolution of the global vehicle stock over 50 years; identify the dominant emission contributors by vehicle and fuel type; and further characterize the age distribution of on-road $CO_2$ emissions. Compared to the publicly available on-road $CO_2$ emissions from previous studies, $CO_2$ emissions in this study have more detailed source categories which are refined into vehicle and fuel type. And with the age distribution simulated by our fleet turnover model, $CO_2$ emissions offered in this study would better support the policy-making of emission mitigation.

--In this work, the database of fuel-, vehicle type-, and age-specific $CO_2$ emissions from global on-road vehicles from 1970 to 2020 is the key achievement. However, the data provided on FigShare is only in *.mat* format. To facilitate more readers to use this dataset, adding a non-proprietary format (e.g., the netCDF file) is recommend.

Data in the format of netCDF file has been provided on FigShare. The data are available as open data at https://doi.org/10.6084/m9.figshare.24548008

--If the air pollution inventory is to be output at the same time with the $CO_2$ emission inventory, will the fleet turnover model and the input data need to be adjusted significantly?

The air pollution inventory has been built recently and the fleet turnover model was not adjusted significantly. The developed modules as well as the input data remain unchanged, two new modules were added to the model to estimate emissions of air pollutants. One added module was judgment of emission standard, in which the output of age distribution simulation was used as input and the proportion of each emission standard stage by vehicle and fuel types in target year and country was output. The other added module was emission factor estimation, in which emission

factors in running, start, and evaporation state of target emission standard stage, vehicle and fuel type was estimated. The output of emission factor estimation was the input of vehicular emission estimation.

**Specific comments:**

- Please align the text formatting, e.g. line spacing is not aligned.

  Text formatting has been aligned.

- Some of the references to figures and equations use abbreviations but some do not. For example, in line 83 using "(Eq. 5)", in line 84 using "(Figure 1)". Please check.

  References to figures and equations have been used uniformly.

- Line 85, "In summary," --> "Specifically,"

  Wording has been modified.

- Line 89, "Then," --> "Third,"

  Wording has been modified.

- Figure 2, Figure S3-4, is there any interannual variation of the performance of the modeled vehicle stock and the age distribution during 1970-2020?

  In general, the performance of the modeled vehicle stock and the age distribution would be better in more recent years as more statistics were available. Taking United State as an example, the relative deviation ratios of vehicle stock ranges from 7% to 17% during 1970-1980, while the range of relative deviation ratios decreases to around $\pm 3\%$ after 1990. As the validation indicator of age distribution, the relative deviation between the simulated vehicle stock based on newly registered vehicles and survival rates and the vehicle stock for PLDVs in United States ranges from 11% to 17% during 1970-1980, while relative deviation ratios after 1990 are between -7% and 10%.

- Figure S12: why $CO_2$ emissions around 1990 are visibly higher than that in adjacent years in in rest of Europe?

  The higher $CO_2$ emissions around 1990 in rest of Europe were mainly from countries of the former Soviet Union. There's an abrupt jump in the national on-road fuel consumption of these countries derived from IEA around 1990, which leads to the visible higher $CO_2$ emissions. The mutation of IEA fuel consumption statistics around 1990 in countries of the former Soviet Union may have been influenced by the collapse of the Soviet Union.

**RC2**

1. What is the difference between the results of this study and the emissions at the sector level? If the difference is within an order of magnitude, authors should consider whether this work is still meaningful.

   Although the difference in total $CO_2$ emissions from global on-road vehicles is within an order of magnitude, source category of emissions in this study is refined into vehicle and fuel type, and age distribution is also offered in our public data at the same time. The fuel-, vehicle type-, and age-specific emissions offered in this study could not be obtained from existing studies and would better support the policy-making of emission mitigation.

2. Since the article has established a model, how did the author validate the model results?

   As the verification of the vehicle stock estimated by our model, we compared them with the statistical vehicle stock (Figure 2(a) and Figure S3). The relative deviation ratios in countries that own top 85% of global vehicles stock were between -28% and 25.6%, and ranges of the relative deviation in rest countries were a bit larger due to the limited availability of statistics. The deviation of the modeled vehicle stock from the statistics in most countries was less than ±25%, especially in the United States, countries in the European Union, China, and India. The relatively good consistency between the modeled and statistical vehicle stock indicates the relatively high reliability of this model.

   To verify the age distribution simulated by our model, survived vehicles calculated by newly registered vehicles and survival rates were compared to the vehicle stock from our integrated vehicle stock database (Figure 2(b) and Figure S4). Except for several years in Argentina and Thailand, the relative deviation ratios of light-duty vehicles during 1970-2020 ranges from -30.9% to 30.8%, heavy-duty vehicles had larger relative deviation ratios which were between -36.5% and 34.9%. The relatively good consistency between the vehicle stock and simulation indicated that the dynamic balance function set up in this study could well model the entry of newly registered vehicles and the retirement of existing vehicles and the estimated age distribution was reliable.

[Figure]

Figure 2: Verification of the modeled vehicle stock in United States, the European Union, China, and India (a) and the age distribution for PLDVs (b) in 2015.

[Figure]

Figure S3. Verification of the modeled vehicle stock in rest countries in 2015.

[Figure]

Figure S4. Verification of the age distribution for CLDVs, buses, and trucks in 2015.

3. The spatiotemporal resolutions of this dataset are too low to apply to other models.

   This study is aimed to offer global on-road $CO_2$ emissions with detailed source category (refined into vehicle and fuel type) and age distribution, the spatiotemporal distribution of emissions will be completed in our follow-up work.

4. Emission factors from the IPCC overestimate CO2 emissions, which increases the uncertainty. If the activity level data in this paper are reliable, where are the differences between the sector level and yours?

   In this study, local emission factors were used in countries where local data was available and emission factors from IPCC were used in countries lack of local studies. Local $CO_2$ emission factors used in this study were lower compared to that from IPCC. Taking China and Europe as an example, $CO_2$ emission factor of diesel vehicles from IPCC was 3186.3g/kg, while in our local references it was 3159.091 g/kg for China and 3140 g/kg for Europe. In this study, 52 to 70% of $CO_2$ emissions were estimated with local emission factors, the rest 30 to 48% were estimated using IPCC emission factors.

   Differences between the sector level and ours mainly lie in the source category of emissions. Source category of $CO_2$ emissions offered in this study is refined into vehicle and fuel type, and age distribution is also offered in our public data at the same time. However, existing $CO_2$ emissions from global on-road vehicles were publicly available, at best, by fuel type.

5. The whole paper describes the results and lacks an analysis to explain why it shows this trend.

   Explanation has been added. According to the ESSD guidelines which require authors give focus to the data and less on its interpretation, the texts do not stretch much.

6. This work can provide a basic dataset for other research; however, they did not provide and discuss the reliability of this work.

   The corresponding uncertainty was calculated in this study to quantify $CO_2$ emission uncertainty. In the uncertainty assessment, uncertainty values of emission factors were derived from EEA, and countries in this study were divided into 12 groups in accordance with IPCC tiered approach and EDGAR to evaluate the uncertainty of activity data. For 15 member countries of European Union (EU15), uncertainty values were obtained from Olivier et al. (2016). For countries belonging to the OECD in 1990

(OECD90), we assumed that they had the lowest uncertainty values. For countries with Economies in Transition of 1990 (EIT90), we assumed that they were more uncertain than OECD90 but less than countries in development (the UNFCCC nonAnnex I). Australia, India, China, Canada, Japan, Russia, Ukraine, and United States did not belong to above four groups, their uncertainty values were obtained from Olivier et al. (2016) and Hong et al. (2017).

The uncertainty in the global on-road $CO_2$ emissions was estimated to range from -7.2% to 8.1%, which is close to the expert judgement suggested value (approximately ±5%) in GPG (2000). It's found that uncertainty in $CO_2$ emissions from on-road vehicles varied significantly among countries and regions. United States and European Union had the lowest uncertainty in the range of -3.8% to 4.0% and -2.9% to 3.0%, respectively, which owes to their sufficient local data. Due to the less-developed statistical systems, Latin Am. + Canada and Middle East + Africa have the largest uncertainty, which ranged from -12.3% to 14.6% and -15.4% to 18.3%, respectively. China's relatively larger uncertainty, with the range of -12.6% to 14.4%, came from the relatively large apparent uncertainties (~15.8%) in oil consumption statistics in China during 1996-2003 (Hong et al., 2017). India had relatively low uncertainty that varies between -4.7% and 5.0% because of the low uncertainty values derived from Janssens-Maenhout et al. (2019) in which India was classified as countries with well-developed statistical systems. It could also be found that uncertainties at regional level decreased over time with the development of statistical systems in more countries. But uncertainty in global on-road $CO_2$ emissions slightly increased during 1970-2020 due to the growing contribution of regions with larger uncertainty to the global total $CO_2$ emissions. Table S4 shows the he corresponding uncertainty (σ) of $CO_2$ emissions for regions

Table S4. The corresponding uncertainty (σ) of $CO_2$ emissions for regions.

| Region | 1970 | 1980 | 1990 | 2000 | 2010 | 2015 | 2020 |
|---|---|---|---|---|---|---|---|
| World | (-5.5%, 6.2%) | (-5.8%, 6.5%) | (-5.8%, 6.5%) | (-6%, 6.7%) | (-7%, 8%) | (-6%, 6.7%) | (-5.9%, 6.6%) |
| United States | (-3.8%, 4%) | (-3.7%, 3.8%) | (-3.5%, 3.6%) | (-3.4%, 3.5%) | (-3.2%, 3.3%) | (-3.2%, 3.3%) | (-3.1%, 3.2%) |
| European Union | (-2.9%, 3%) | (-2.7%, 2.8%) | (-2.5%, 2.6%) | (-2.3%, 2.4%) | (-2.5%, 2.7%) | (-2.5%, 2.6%) | (-2.7%, 2.8%) |
| China | (-6.8%, 7.3%) | (-7.1%, 7.7%) | (-7.5%, 8.2%) | (-7.9%, 8.5%) | (-11.8%, 13.4%) | (-1.4%, 1.4%) | (-1.2%, 1.2%) |
| India | (-4.7%, 5%) | (-4.5%, 4.7%) | (-4.3%, 4.5%) | (-4.2%, 4.4%) | (-4.1%, 4.2%) | (-3.9%, 4.1%) | (-3.9%, 4%) |
| Latin Am. + Canada | (-12.2%, 14.6%) | (-11.9%, 14.1%) | (-12.1%, 14.3%) | (-12.2%, 14.3%) | (-11.9%, 13.9%) | (-11.8%, 13.7%) | (-11.3%, 13.1%) |

| Middle East + Africa | (-15.4%, 18.3%) | (-15.1%, 18%) | (-14.3%, 16.8%) | (-13.9%, 16.2%) | (-13.6%, 15.8%) | (-13.2%, 15.3%) | (-12.8%, 14.9%) |
|---|---|---|---|---|---|---|---|
| Rest of Asia | (-7.3%, 8.6%) | (-7%, 8.2%) | (-7.1%, 8.3%) | (-7.4%, 8.6%) | (-8.5%, 9.8%) | (-9.1%, 10.5%) | (-9.3%, 10.7%) |
| Rest of Europe | (-6.4%, 7.1%) | (-5.9%, 6.5%) | (-5.4%, 5.8%) | (-4.7%, 5.1%) | (-4.9%, 5.3%) | (-4.9%, 5.3%) | (-4.9%, 5.3%) |
| Rest of world | (-4.5%, 4.8%) | (-4.4%, 4.8%) | (-4.4%, 4.8%) | (-4.3%, 4.7%) | (-4.5%, 5%) | (-4.3%, 4.7%) | (-4.2%, 4.6%) |

7. The content of this paper is thin and slim, authors should provide at least one application of this dataset (such as Earth system models, atmospheric chemistry and transport models, and integrated assessment).

This study focuses on the setup and evaluation of a global vehicle emission model and dataset. This is the general framework for emission inventory studies. Unlike existing global inventories, our study built a global fleet turnover model driven by comprehensive, harmonized datasets from multiple sources. This approach enhances the accuracy of $CO_2$ emission estimates and provides detailed, fuel-, vehicle type-, and age-specific $CO_2$ emissions data from global on-road vehicles, which are currently unavailable in other databases. This study encompasses extensive dataset processing, development, and evaluation, all of which are time- and labor-intensive. We agree that the atmospheric modeling community will greatly benefit from our high-resolution emission dataset. Although we intend to conduct modeling studies in the future, this initial paper from our group concentrates on the development and evaluation of the emission dataset, reserving the application of these datasets for subsequent research.

8. Although the writing seems good, there are problems in this paper:

The text in this paper has been edited according to comments.

9. L247: 2020 appears at the end of the sentence and the beginning of another sentence;

The expression has been changed as:

… in India in 2020. The majority of vehicles in the European Union in 2020 were still PLDVs, for which the proportion was 79%, …

10. L247-249: The subject of the before and after inflection is inconsistent;

The expression has been changed as:

…, but the dominant vehicle type in United States has changed from PLDVs to CLDVs and CLDVs accounted for 50% of the local vehicle stock.

11. L249-251:The first half of this sentence is ambiguous.

This sentence has been changed as:

As the dominant position of developed countries in global vehicle stock replaced by developing countries during the 1970-2020 period (Figure S6), the share of MCs in the global vehicle stock increased accordingly to 32%, and the proportion of PLDVs decreased to 50% in 2020.

12. The dashed line in Figure 1 exceeds the boundary.

Figure 1 has been changed as:

[Figure]

**List of relevant changes**

1. Adding the motivation of this work as well as the limitation of current inventories (L58-61):

*Compared to the publicly available on-road CO2 emissions from previous studies, CO2 emissions in this study have more detailed source categories which are refined into vehicle and fuel type. And with the age distribution simulated by our fleet turnover model, CO2 emissions offered in this study would better support the policy-making of emission mitigation.*

2. Updating data link as:*https://doi.org/10.6084/m9.figshare.24548008*

3. Changing "*Eg.*" to "*Equation*" (L76, 80, 84, 87, 91, 95,97, and 226).

4. Changing "*Fig.*" to "*Figure*" (L100, 145, 271, 301, 318, 336, and 362).

5. Changing "*In summary,*" to "*Specifically,*" (L89).

6. Changing "*Then,*" to "*Third,*" (L93).

7. Aligning the line spacing to 1.5 times space, consistent with the ESSD template.

8. Changing expression according to commets 9 and 10 in RC2, and adding explaination of data trend according to commet 5 in RC2 (L251-256):

*…, MCs with the proportion of 78% the vehicle stock in 2020 were still the most frequently used vehicles in India, benefiting by the local warm climateMCs were still the most frequently used vehicles, accounting for 78% of the vehicle stock in India in 2020. In 2020, tThe majority of vehicles in the European Union in 2020 were still PLDVs, …*

9. Changing expression according to commet 11 in RC2 (L256-259):

*As the dominant position of developed countries in global vehicle stock replaced by developing countries during the 1970-2020 period (Figure S6), …*

10. Changing the dashed line in Figure 1 (L98):